# Impact of Diabetes Duration on Clinical Outcome in Patients Receiving Rotational Atherectomy in Calcified Lesions in Korea—Results from ROCK Registry

**DOI:** 10.3390/life12070993

**Published:** 2022-07-04

**Authors:** Jin Jung, Sung-Ho Her, Kyusup Lee, Ji-Hoon Jung, Ki-Dong Yoo, Keon-Woong Moon, Donggyu Moon, Su-Nam Lee, Won-Young Jang, Ik-Jun Choi, Jae-Hwan Lee, Jang-Hoon Lee, Sang-Rok Lee, Seung-Whan Lee, Kyeong-Ho Yun, Hyun-Jong Lee

**Affiliations:** 1Department of Cardiology, St. Vincent’s Hospital, College of Medicine, The Catholic University of Korea, Seoul 16247, Korea; colaking@naver.com (J.J.); yookd@catholic.ac.kr (K.-D.Y.); eheart@catholic.ac.kr (K.-W.M.); babaheesu@gmail.com (D.M.); yellow-night@hanmail.net (S.-N.L.); raph83@naver.com (W.-Y.J.); 2Department of Cardiology, Daejeon St. Mary’s Hospital, College of Medicine, The Catholic University of Korea, Seoul 34943, Korea; 3Korea Institute of Toxicology, Daejeon 34114, Korea; jihoon.jung@kitox.re.kr; 4Department of Cardiology, Incheon St. Mary’s Hospital, College of Medicine, The Catholic University of Korea, Incheon 21431, Korea; mrfasthand@catholic.ac.kr; 5Department of Cardiology in Internal Medicine, Chungnam National University School of Medicine, Daejeon 35015, Korea; myheart@cnuh.co.kr; 6Department of Internal Medicine, Kyungpook National University Hospital, Daegu 41944, Korea; ljhmh75@knu.ac.kr; 7Department of Cardiology, Chonbuk National University Hospital, Jeonju 54907, Korea; medorche@naver.com; 8Department of Cardiology, Asan Medical Center, University of Ulsan College of Medicine, Seoul 05505, Korea; seungwlee@amc.seoul.kr; 9Department of Cardiovascular Medicine, Regional Cardiocerebrovascular Center, Wonkwang University Hospital, Iksan 54538, Korea; ards7210@wonkwang.ac.kr; 10Department of Internal Medicine, Sejong General Hospital, Bucheon 14754, Korea; untouchables00@hanmail.net

**Keywords:** coronary artery calcification, diabetic duration, rotational atherectomy, clinical outcome

## Abstract

There are limited data regarding the clinical impact of diabetes duration for patients with heavy calcified coronary lesions. We sought to determine the clinical impact of diabetes duration on clinical outcomes in patients with heavily calcified lesions who required rotational atherectomy during percutaneous coronary intervention (PCI). A total of 540 diabetic patients (583 lesions) were enrolled between January 2010 and October 2019. Patients were classified into three subgroups: patients with no diabetes mellitus (non-DM), shorter duration (S-DM), and longer duration (L-DM), of which duration was divided at 10 years. During 18 months of follow-up-duration, diabetes duration was significantly associated with the primary outcome. The incidence rate of target-vessel failure (TVF), the primary outcome, was significantly higher in the L-DM group compared with non-DM or S-DM. Among secondary outcomes, any repeat revascularization (RR) was frequently observed in the L-DM compared with other groups. In multivariate analysis, the risk of TVF and any RR was 1.9 times and 2.4 times higher in L-DM than in non-DM, respectively. This study firstly demonstrated that there is an association between a longer DM duration and poor clinical outcomes in patients with severe calcified CAD after PCI. More careful monitoring for recurrence is needed during follow-up in those patients.

## 1. Introduction

Diabetes mellitus (DM) is a well-known risk factor for coronary artery disease (CAD) [1] and is also associated with coronary artery calcifications (CACs), greater atherosclerosis burden, and multivessel disease [2,3], which consequently results in poor clinical outcomes following percutaneous coronary intervention (PCI) [4,5,6].

Since DM is a chronic, progressive disease leading to micro- or macrovessel damage [7], the duration of diabetes (DM duration) is also associated with clinical outcomes in patients with CAD. In previous studies, DM duration was independently associated with cardiovascular mortality and increased the risk of coronary heart disease death [8,9]. Since a long period (>10.5 years) of prevalence of DM is significantly associated with adverse cardiovascular events, it was suggested to consider routine CAD screening examination in these patients [10]. There were several studies comparing CAD mortality and clinical outcomes following PCI, including rotational atherectomy (RA), between diabetes and non-diabetic patients [6,11,12]. However, the clinical impact of DM duration is not well-known, and there have been no studies describing the relationship between DM duration and clinical outcome after PCI, especially in heavy calcified lesions. Therefore, we sought to determine the clinical impact of DM duration on clinical outcomes in patients with heavy CAC lesions underwent PCI using RA.

Based on the 10-year prevalence period, we compared the clinical outcome and procedure details during RA.

## 2. Materials and Methods

### 2.1. Study Design and Population

The study population consisted of 540 patients (583 lesions) with heavily calcified CAD who received PCI using RA from January 2010 to October 2019 at 9 tertiary centers in Korea within the ROCK registry approved by the institutional review board of each hospital. Data were collected at enrolled centers using a standardized case report form to record clinical characteristics, demographic characteristics, procedural data, and follow-up data. Follow-up data were collected up to 18 months based on clinical records and on physician or patient interviews at the time of registry enrollment. This study was approved by the regional ethics committee for each participating hospital, and all patients provided their written informed consent to the use of medical data for the registry study.

To determine the presence and duration of diabetes, we reviewed the medical records; otherwise, they were self-reported. DM duration was calculated as the difference between age and age of onset of diabetes. Patients whose duration of diabetes was not investigated were excluded from the study

Patients divided into three subgroups based on 10 years duration. The reason for dividing the groups at 10 years is that a type 2 DM duration of 10 years is known to be the point at which beta cell loss becomes irreversible and cannot be restored through the achievement of glycemic control [13]. This process leads to high risk for the development of macro- and microvascular complications [14]. In addition, we refer to previous studies in which a duration of 10 or more years predicts significant CAD, increases the risk of cardiovascular disease, and is associated with higher adverse cardiovascular events [10]. The flow chart is displayed in Figure 1. The baseline characteristics of the study patients and clinical outcomes were compared between the three groups.

### 2.2. RA Procedure

Procedure details including RA standard technique, RA system, and treatment strategy were same as previously published report [15]. During follow-up, patient management, including medical treatment such as peri-procedural anticoagulation and antiplatelet therapy, was performed in accordance with established standards of care and accepted guidelines.

### 2.3. Clinical Outcomes

The primary endpoint was the composite rate of target-vessel failure (TVF), defined as target-vessel revascularization (TVR), target-vessel spontaneous myocardial infarction (TVMI), or cardiac death. The secondary endpoints included all-cause death, cardiac death, TVMI, any MI, any revascularization, and TVR. Technical/procedural success, in-hospital events, or peri-procedural complication were also investigated. The definition of outcomes was the same as in the previously published report mentioned above [15].

Especially, diabetes was defined as either a previously diagnosed DM or newly diagnosed DM by applying the 2010 criteria of the American Diabetes Association. According to this definition, subjects with fasting glucose ≥126 mg/dL and/or glycated hemoglobin ≥6.5% and/or post-challenge glucose (glucose at 2 h after a 75 g oral glucose load) ≥200 mg/dL were newly diagnosed with DM [16]. Peri-procedural MI was defined as a peak rise of the creatine kinase-myocardial band 10 times higher than the upper limit of normal within 48 h after PCI. Cerebrovascular accident (CVA) was defined as a focal neurological defect of central origin lasting more than 24 h and confirmed by a neurologist and imaging. Chronic kidney disease (CKD) was defined as a calculated glomerular filtration rate <60 mL/min/1.73 m^2^ by the Modification of Renal Diet (MDRD) equation from baseline serum creatinine [17]. Contrast-induced nephropathy (CIN) after the procedure was defined as an impairment of the function of the kidney, measured as either a 0.5 mg/dL rise in absolute serum creatinine level or a 25% rise in serum creatinine compared to baseline level within 48–72 h after the procedure.

### 2.4. Statistical Analysis

Continuous variables are presented as the median and interquartile range or mean ± standard deviation using Student’s *t*-test. Categorical variables were expressed by numbers and percentages. Differences between the groups, categorized according to the DM duration, were compared using analysis of variance (ANOVA) or the Kruskal–Wallis test for continuous variables, and the chi-square test or Fisher’s exact test for categorical variables as appropriate. Post hoc tests were performed using ANOVA with the Tukey method or the Kruskal–Wallis test with Bonferroni correction. Univariable and multivariable Cox regression analyses were performed to analyze the impact of DM duration on clinical outcomes. The hazard ratio (HR) and 95% confidence interval (CI) were also calculated. For multivariate analysis, confounding factors were age, sex, hyperlipidemia, CKD, dialysis, CVA, PVD, multivessel disease (MVD), Hb, total cholesterol, LDL cholesterol, HbA1c, and contrast-induced nephropathy. Event rates were determined using Kaplan–Meier method in time-to-first-event analyses and were compared by the log-rank test. For subgroup analysis, Cox regression analysis was performed and visualized by forest plots. A *p* value < 0.05 was regarded statistically significant. All statistical analyses were performed using Statistical Analysis Software (SAS, version 9.2, SAS Institute, Cary, NC, USA).

## 3. Results

### 3.1. Baseline Characteristics

Patients were divided into three groups according to the DM duration. Among a total of 583 lesions, 133 lesions of which DM duration was not investigated were excluded from the study. The remaining 450 lesions were analyzed and divided into three subgroups based on 10 years of DM duration: (1) non-DM group (251 lesions), (2) a shorter duration DM group (65 lesions) with DM duration of <10 years (S-DM), and (3) a longer duration DM group (134 lesions) with DM duration of ≥10 years (L-DM).

Table 1 presents a comparison of baseline characteristics between the non-DM, S-DM, and L-DM groups. There was no significant difference in demographic variables among the three groups except for proportion of gender, CKD, and history of hyperlipidemia, CVA, and PVD, which were more frequently observed in the diabetic groups than non-DM group. Especially, the proportions of CKD and dialysis, which were known as risk factors for poor prognosis, were statistically significant and gradually increased in order of non-DM, S-DM, and L-DM (CKD, 27 [0.8%] vs. 8 [12.3%] vs. 37 [27.6%], *p* < 0.001; dialysis, 15 [6.0%] vs. 5 [7.7%] vs. 20 [14.9%], *p* = 0.012). Moreover, the level of lipid profile was different between non-DM versus S-DM or L-DM, which showed significantly lower in the diabetic groups than in the non-DM group. The level of HbA1c level increased gradually according to the DM duration (5.8 ± 0.5 vs. 7.0 ± 1.3 vs. 7.6 ± 1.9, *p* < 0.001, respectively). MVD was lower in the no DM group than in the diabetic group, but there was no difference between the S-DM and L-DM groups (189 [75.3%] vs. 56 [86.2%] vs. 115 [85.8%], *p* = 0.02, respectively; Table 2).

### 3.2. Procedural Details, In-Hospital Events, and Procedure Complications

Procedural details including procedure time, mean stent diameter, total stent length, and technical/procedure success rates were similar among the three groups (Table 2). The incidence rates of in-hospital MACCEs were also similar among the three groups (Table 3). However, CIN occurred more frequently in the L-DM group compared with non-DM or S-DM group.

### 3.3. Mid-Term Clinical Outcomes

During median follow-up duration of 18 months, the incidence rate of TVF, primary outcome, was significantly higher in the L-DM group compared with non-DM or S-DM groups (non-DM, 30 [12.0%] vs. S-DM, 9 [13.9%] vs. L-DM, 29 [21.6%]; *p* = 0.039). Among secondary outcomes, any repeat revascularization was frequently observed in the L-DM compared with other groups (non-DM, 19 [7.6%] vs. S-DM, 6 [9.2%] vs. L-DM, 21 [15.7%]; *p* = 0.042; Table 4) (Figure 2).

In multivariate analysis, L-DM group showed significantly poorer clinical outcomes than non-DM group in terms of TVF and RR (TVF: hazard ratio [HR] 1.86, 95% confidence interval [CI] 1.04–3.34, *p* = 0.037; RR: HR 2.40, 95% CI 1.17–4.89, *p* = 0.017), while S-DM group did not show statistical significance (TVF: HR 1.04, 95% CI 0.45–2.44, *p* = 0.92; RR: HR 1.27, 95% CI 0.47–3.46, *p* = 0.64) (Table 5).

In subgroup analysis comparing the primary outcome for non-DM and L-DM groups, the sex difference was observed that the relative risk of TVF was higher in the L-DM group compared with the non-DM group in females, which was not observed in males (p_interaction_ = 0.033). However, there was no risk difference in the subgroups with CKD, dialysis, or lesions of chronic total occlusion (Figure 3).

## 4. Discussion

The major findings from the present analysis are as follows: (1) patients with longer DM duration, especially more than 10 years, showed poorer clinical outcomes in terms of TVF and RR than non-diabetic patients or those with shorter DM duration even after revascularization; (2) a long DM duration did not affect the procedural details, in-hospital and procedural outcomes except for CIN during PCI using RA.

This is the first study to examine the associations between DM duration and clinical outcomes in a Korean population with CAD. Our registry is the largest, all-comer, multi-center registry, including CAD patients with advanced atherosclerosis in Korea. All patients received PCI with drug-eluting stents (DESs), especially second-generation DES, except for one patient with a first-generation DES, which reflect the current revascularization strategy for significant CAD [18,19].

Additionally, all patients underwent RA during PCI, which meant that the majority of patients in this study population might have significant CAD lesions with severe calcification. Using this registry, our study showed the novel results that demonstrated the clinical impact of DM duration on clinical outcomes in advanced CAD after revascularization. Indeed, there have been several studies demonstrating poor clinical outcomes of DM duration in CAD patients [8,20]. However, those studies were designed as cohort or epidemiological studies with relatively healthy patients, while the majority of patients in this study received treatment at tertiary or specialized cardiology centers and may have more severe disease compared with community-treated patients. Therefore, this study has strengths over prior studies that have examined the relation between DM duration and clinical outcomes in revascularized CAD patients with advanced atherosclerosis and calcification.

It is well-known that DM is an independent predictor of adverse clinical outcome in CAD patients after revascularization [5,6]. Our study also showed that patients with a longer duration (≥10 years) of DM showed poorer clinical outcomes regarding TVF and RR than non-DM and shorter duration DM (<10 years) groups. The risk of TVF and any revascularization was 1.863 times and 2.395 times higher in L-DM than in non-DM, respectively. The plausible mechanisms are as follows: (1) CAD severity and the extent atherosclerosis is getting worse during prolonged diabetes [2,3]. (2) DM duration had a strong correlation with the nature of unfavorable coronary artery lesions. Vulnerable plaque including lipid-rich plaque and thin cap fibroatheroma and plaque rupture were frequently observed in the long DM duration (≥10 year) group compared to shorter duration DM or non-DM group [21,22]. (3) CAC can be promoted by multifocal factors of the hormonal and physiological abnormalities associated with DM, including oxidative stress, endothelial dysfunction, and increased inflammatory cytokine production [23,24]. These changes were enhanced by prolonged DM with poor glycemic control [25], leading to adverse procedural complications and long-term clinical outcomes [26,27,28].

However, the incidence rate of clinical outcomes in the shorter DM duration group was numerically higher but did not achieve statistical significance compared to the non-DM group in this study. In our registry, the duration of S-DM group was relatively short (median 5 [IQR 0.17–7.33] years), including 20 (30.8%) patients who were diagnosed as DM within 1 year. Therefore, the S-DM group may include diabetic patients without complications. Even though the presence of DM itself is associated with poor clinical outcomes as mentioned above [5,6], adverse clinical events are usually prevented by revascularization with optimization, which is already known as an important protective predictor [29]. Several previous studies have been in line with this result in patients with CAD [30,31].

Our study showed similar results regarding the procedural details as well as in-hospital and procedural outcomes, except for CIN. CIN occurred more frequently in the longer DM duration group. Prolonged DM is associated with microvascular complication including microalbuminuria [20,32,33]. In this context, we could understand that CKD was frequently observed in the L-DM group. This would be a plausible explanation of difference in CIN occurrence after procedure.

Operators are usually more careful with patients with diffuse narrowing CAD during RA and worry about distal embolization, leading to flow compromising and peri-procedure MI. Since DM may affect the coronary vessel more diffusely narrowing including capillaries [34] and longer DM duration reduces myocardial blood flow in remote myocardium [35], diabetic patients would have a greater chance of suffering peri-procedure MI. However, the incidence rates of peri-procedure MI were similar among the three groups in our study. According to the results, one of our messages is that, we may not be hesitant with RA, even in longer duration of diabetic patients in terms of slow or no reflow.

In subgroup analysis, the present study was in agreement with several previous studies that showed that the relative risk of vascular disease associated with DM was substantially higher in women than men [36,37]. Although the mechanisms were not fully identified, it may be associated with hormonal and storage patterns of adipose tissue differences [38,39].

## 5. Study Limitation

This study was based on a nonrandomized registry with inherent methodological limitations. Thus, there is a possibility of selection bias. Second, the number of study participants and events was relatively small, limiting the statistical power of our multivariate analysis. Third, the proportion of the S-DM group was relatively small (14%). Therefore, caution is necessary when interpretating our results.

## 6. Conclusions

This study firstly demonstrated that there is an association between a longer DM duration (≥10 years) and poor clinical outcomes in patients with severe calcified CAD lesions after PCI, especially in terms of TVF and RR. We should be more careful about recurrence during follow-up in those patients.

## Figures and Tables

**Figure 1 life-12-00993-f001:**
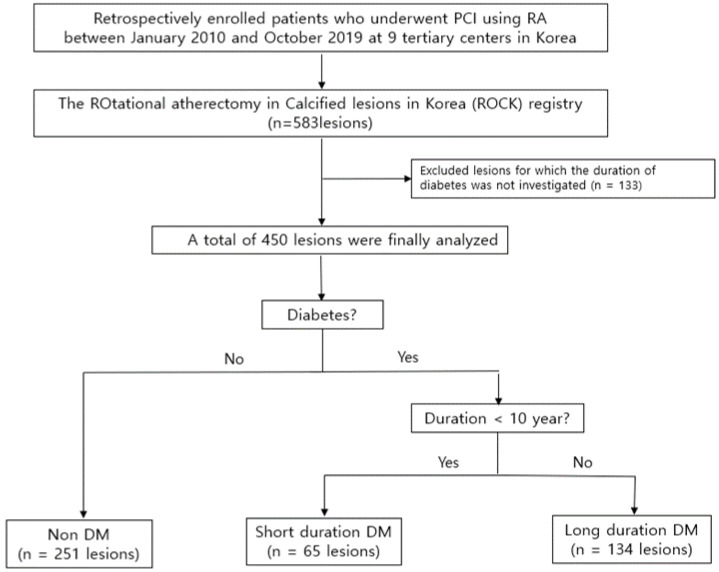
Study population flow chart. PCI, percutaneous coronary intervention; RA, rotational atherectomy; DM, diabetes mellitus.

**Figure 2 life-12-00993-f002:**
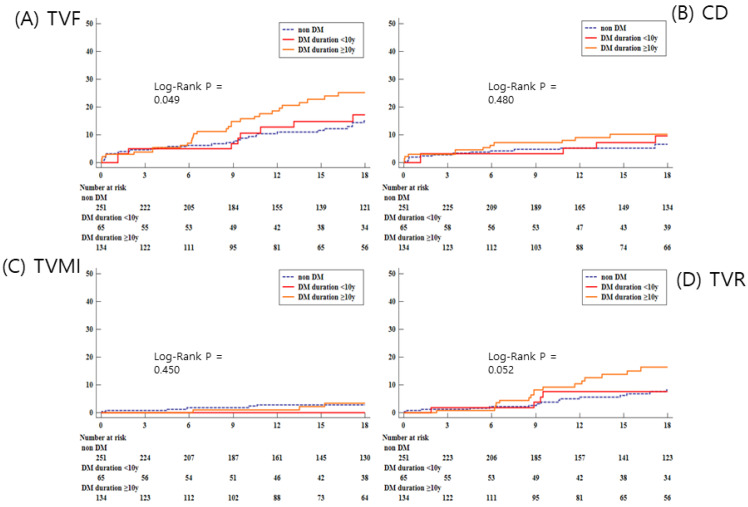
Kaplan–Meier curve for clinical outcomes during follow up. DM, diabetes mellitus; TVF, target vessel failure; CD, cardiac death; TVMI, target vessel myocardial infarction; TVR, target vessel revascularization.

**Figure 3 life-12-00993-f003:**
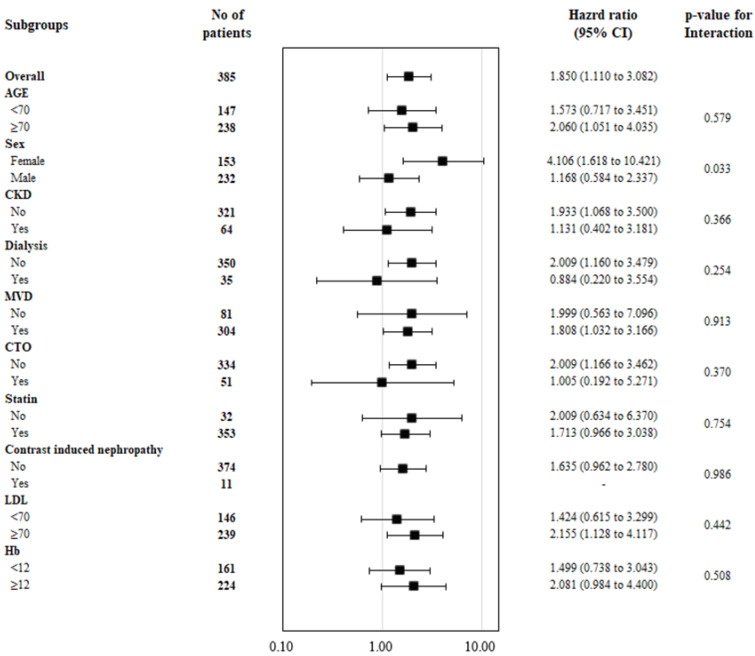
Forest plot in subgroup analysis comparing the target vessel failure of Non-DM and Long-duration DM. DM, diabetes mellitus; CKD, chronic kidney damage; MVD, multivessel disease; CTO, chronic total occlusion; CI, confidence interval.

**Table 1 life-12-00993-t001:** Baseline characteristics.

	Non-DM (n = 251)	S-DM (n = 65)	L-DM (n = 134)	*p*-Value	Post-Hoc
DM duration, years		5 (0.17–7.33)	20 (13.75–26)		
Age, years	71.8 ± 10.9	71.9 ± 9.7	70.5 ± 9.0	0.50	
Sex				0.018	1 > 3
Male	164 (65.3)	37 (56.9)	68 (50.8)		
Female	87 (34.7)	28 (43.1)	66 (49.2)		
Smoking	53 (21.1)	14 (21.5)	26 (19.4)	0.91	
BMI	24.18 ± 4.23	24.40 ± 3.26	24.03 ± 3.51	0.82	
HTN	182 (72.5)	52 (80.0)	106 (79.1)	0.24	
Hyperlipidemia	95 (37.9)	40 (61.5)	64 (47.8)	0.002	1 < 2
CKD	27 (10.8)	8 (12.3)	37 (27.6)	<0.001	1, 2 < 3
Dialysis	15 (6.0)	5 (7.7)	20 (14.9)	0.012	1 < 3
Previous PCI	54 (21.5)	22 (33.9)	38 (28.4)	0.08	
Previous CABG	9 (3.6)	6 (9.2)	8 (6.0)	0.16	
Previous MI	32 (12.8)	8 (12.3)	12 (9.0)	0.53	
CVA	29 (11.6)	16 (24.6)	20 (14.9)	0.028	1 < 2
PVD	12 (4.8)	9 (13.9)	16 (11.9)	0.011	1 < 2, 3
Chronic lung disease	19 (7.6)	8 (12.3)	5 (3.7)	0.08	
Heart failure	32 (12.8)	16 (24.6)	22 (16.4)	0.06	
Atrial fibrillation	23 (9.2)	11 (16.9)	10 (7.5)	0.10	
Clinical diagnosis				0.66	
Stable angina,	100 (39.8)	24 (36.9)	58 (43.3)		
Acute coronary syndrome	151 (60.2)	41 (63.1)	76 (56.7)		
HbA1C	5.8 ± 0.5	7.0 ± 1.3	7.6 ± 1.9	<0.001	1 < 2 < 3
Total cholesterol	152.5 ± 41.7	139.0 ± 37.6	138.1 ± 34.5	0.001	1 > 2, 3
LDL cholesterol	90.8 ± 36.2	73.4 ± 28.1	76.8 ± 30.2	<0.001	1 > 2, 3
HDL cholesterol	48.5 ± 15.3	43.8 ± 13.5	43.3 ± 14.1	0.003	1 > 2, 3
Triglyceride	118.9 ± 78.0	140.7 ± 70.3	124.1 ± 73.8	0.15	

DM, diabetes mellitus; S-DM, shorter duration of diabetes mellitus; L-DM, longer duration of diabetes mellitus; BMI, body mass index; HTN, hypertension; CKD, chronic kidney disease; PCI, percutaneous coronary intervention; CABG, coronary artery bypass graft; MI, myocardial infarction; CVA, cerebrovascular accident; PVD, peripheral vascular disease; HbA1c, glycated hemoglobin.

**Table 2 life-12-00993-t002:** Baseline angiographic characteristics and procedural details.

	Non-DM (n = 251)	S-DM (n = 65)	L-DM (n = 134)	*p*-Value
Lesion classification				0.22
B1, n (%)	11 (4.4)	1 (1.5)	4 (3.0)	
B2, n (%)	23 (9.2)	2 (3.1)	16 (11.9)	
C, n (%)	217 (86.5)	62 (95.4)	114 (85.1)	
MVD, n (%)	189 (75.3)	56 (86.2)	115 (85.8)	0.02
CTO, n (%)	34 (13.6)	4 (6.2)	17 (12.7)	0.26
Pre EF	54.80 ± 13.08	52.98 ± 13.24	51.90 ± 14.77	0.13
Procedure time, min	83.03 ± 49.32	82.10 ± 40.84	74.39 ± 55.24	0.26
Mean stent diameter, mm	3.01 ± 0.38	3.01 ± 0.39	2.96 ± 0.38	0.38
Total number of stent	2.35 ± 1.15	2.19 ± 1.03	2.55 ± 1.26	0.11
Total stent length, mm	67.09 ± 32.72	60.31 ± 28.42	72.18 ± 39.17	0.08
Number of burr, mm	1.19 ± 0.45	1.25 ± 0.44	1.15 ± 0.36	0.32
Technical success, n (%)	243 (96.8)	61 (93.9)	128 (95.5)	0.52
Procedure success, n (%)	233 (92.8)	60 (92.3)	131 (97.8)	0.11

DM, diabetes mellitus; S-DM, shorter duration of diabetes mellitus; L-DM, longer duration of diabetes mellitus; MVD, multivessel disease; CTO, chronic total occlusion; EF, ejection fraction.

**Table 3 life-12-00993-t003:** In-hospital major adverse cardiac and cerebral events and procedural complications.

	Non-DM (n = 251)	S-DM (n = 65)	L-DM (n = 134)	*p*-Value
In-hospital MACCEs	35 (13.9)	5 (7.7)	15 (11.2)	0.36
In-hospital death	6 (2.4)	1 (1.5)	4 (3.0)	0.82
Urgent revascularization	4 (1.6)	0 (0.0)	4 (3.0)	0.31
In-hospital stroke	2 (0.8)	0 (0.0)	0 (0.0)	0.45
Peri-procedure MI	26 (10.4)	5 (7.7)	8 (6.0)	0.33
Procedural Complications				
Severe coronary dissection *	35 (13.9)	10 (15.4)	19 (14.2)	0.96
Temporary pacemaker during procedure	7 (2.8)	5 (7.7)	5 (3.7)	0.18
Coronary perforation	6 (2.4)	1 (1.5)	1 (0.8)	0.50
Contrast-Induced Nephropathy	4 (1.6)	0 (0.0)	7 (5.2)	0.035
In-hospital bleeding	11 (4.4)	6 (9.2)	8 (6.0)	0.31

***** Type D, E, or F defined from The National Heart, Lung, and Blood Institute (NHLBI) classification system. DM, diabetes mellitus; S-DM, shorter duration of diabetes mellitus; L-DM, longer duration of diabetes mellitus MACCE, major adverse cardiac and cerebral event; MI, myocardial infarction.

**Table 4 life-12-00993-t004:** Clinical outcomes.

	Non-DM (n = 251)	S-DM (n = 65)	L-DM (n = 134)	*p*-Value
TVF, n (%)	30 (12.0)	9 (13.9)	29 (21.6)	0.039
All cause death, n (%)	19 (7.6)	8 (12.3)	12 (9.0)	0.48
Cardiac death, n (%)	14 (5.6)	5 (7.7)	12 (9.0)	0.44
Any myocardial infarction, n (%)	10 (4.0)	0 (0.0)	5 (3.7)	0.27
Target-vessel MI, n (%)	6 (2.4)	0 (0.0)	3 (2.2)	0.46
Any repeat revascularization, n (%)	19 (7.6)	6 (9.2)	21 (15.7)	0.042
TVR, n (%)	15 (6.0)	4 (6.2)	17 (12.7)	0.06

DM, diabetes mellitus; S-DM, shorter duration of diabetes mellitus; L-DM, longer duration of diabetes mellitus; TVF, target vessel failure; MI, myocardial infarction; TVR, target vessel revascularization.

**Table 5 life-12-00993-t005:** Univariable and multivariable Cox regression analysis of clinical outcomes.

		Univariable			Multivariable **		
		HR	95% CI	*p*-Value	HR	95% CI	*p*-Value
Target vessel failure	No DM	1.00			1.00		
	S-DM	1.13	0.54–2.38	0.75	1.04	0.45–2.44	0.92
	L-DM	1.85	1.11–3.01	0.018	1.86	1.04–3.34	0.037
All cause death	No DM	1.00			1.00		
	S-DM	1.55	0.68–3.54	0.30	0.77	0.25–2.35	0.65
	L-DM	1.18	0.57–2.43	0.66	0.98	0.44–2.21	0.97
Cardiac death	No DM	1.00			1.00		
	S-DM	1.32	0.47–3.66	0.60	0.66	0.16–2.67	0.56
	L-DM	1.60	0.74–3.46	0.23	1.67	0.69–4.04	0.26
Myocardial infarction	No DM	1.00			1.00		
	S-DM	-	-	-	-	-	-
	L-DM	0.91	0.31–2.67	0.87	0.85	0.23–3.21	0.81
Target vessel MI	No DM	1.00			1.00		
	S-DM	-	-	-	-	-	-
	L-DM	0.92	0.23–3.69	0.91	0.82	0.12–5.12	0.85
Any revascularization	No DM	1.00			1.00		
	S-DM	1.22	0.49–3.05	0.68	1.27	0.47–3.46	0.64
	L-DM	2.20	1.18–4.10	0.013	2.40	1.17–4.89	0.017
TVR	No DM	1.00			1.00		
	S-DM	1.01	0.33–3.03	0.99	0.87	0.26–2.98	0.83
	L-DM	2.21	1.10–4.43	0.025	2.16	0.94–4.94	0.07

** adjusted by age, sex, hyperlipidemia, CKD, dialysis, CVA, PVD, MVD, Hb, Total cholesterol, LDL cholesterol, HDL cholesterol, Hba1c, Contrast-induced nephropathy. DM, diabetes mellitus; S-DM, shorter duration of diabetes mellitus; L-DM, longer duration of diabetes mellitus; HR, hazard ratio; CI, confidence interval; MI, myocardial infarction; TVR, target vessel revascularization.

## Data Availability

Not applicable.

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
