# Peer review of "Impact of Diabetes Duration on Clinical Outcome in Patients Receiving Rotational Atherectomy in Calcified Lesions in Korea—Results from ROCK Registry"

_life, 2022, doi:10.3390/life12070993_

Round 1

Reviewer 1 Report

Dear Authors,

First, I would like to congratulate you for the excellent article produced. In addition, I noticed that you have already published three others on the topic developed in the evaluated article (doi: 10.1007/s00380-021-01849-4; doi: 10.3390/medicina57070694 and doi: 10.5144/0256-4947.2021.191), which demonstrates your expertise on the subject, as well as demonstrating the importance of the results obtained from carrying out the study developed. However, after reading the other articles published by you, I was able to observe that there are several excerpts very similar to parts of the texts of previously published articles, so I ask you to please make adjustments to the text so that this degree of similarity found is resolved and that the article produced by you can be published.

Kind regards,

Author Response

Response to Reviewer Comments

Point 1: First, I would like to congratulate you for the excellent article produced. In addition, I noticed that you have already published three others on the topic developed in the evaluated article (doi: 10.1007/s00380-021-01849-4; doi: 10.3390/medicina57070694 and doi: 10.5144/0256-4947.2021.191), which demonstrates your expertise on the subject, as well as demonstrating the importance of the results obtained from carrying out the study developed. However, after reading the other articles published by you, I was able to observe that there are several excerpts very similar to parts of the texts of previously published articles, so I ask you to please make adjustments to the text so that this degree of similarity found is resolved and that the article produced by you can be published.

Response 1:. We appreciate your valuable comment. The articles published based on the ROCK registry you mentioned had different topics from this study. However, especially in the Materials and Method section, the composition of the study population, the definition of parameters for clinical outcome, and the procedures related contents were inevitably similar. However, in order to resolve the degree of similarity, we have modified the text in agreement with your comments that text adjustments were necessary

Original text

The study population consisted of 540 patients (583 lesions) with calcified CAD who underwent PCI using RA between January 2010 and October 2019 at 9 tertiary centers

Modified text

The study population consisted of 540 patients (583 lesions) with heavily calcified CAD who received PCI using RA from January 2010 to October 2019 at 9 tertiary centers

Original text

Data were collected at enrolled centers using a standardized case report form to record demographic characteristics, clinical characteristics, procedural data and follow-up data.

Modified text

Data were collected at enrolled centers using a standardized case report form to record clinical characteristics, demographic characteristics, procedural data and follow-up data.

Original text

Follow-up data were collected up to 18 months based on medical records and on physician or patient interviews at the time of registry enrollment

Modified text

Follow-up data were collected up to 18 months based on clinical records and on physician or patient interviews at the time of registry enrollment

This study was approved by the regional ethics committee for each participating hospital, and all patients provided their written informed consent to the use of medical data for the registry study

Original text

The treatment strategy, including decisions regarding burr size during the procedure, was dependent on the discretion of the attending operators. All procedures were guided by standard techniques and management. All RA procedures were performed using the RotablatorTM RA system (Boston Scientific, Marlborough, MA, USA). Antiplatelet therapy and peri-procedural anticoagulation were performed according to the accepted guidelines.

Modified text

Procedure details including RA standard technique, RA system and treatment strategy were same as previously published report [15].

  1. Lee, K.; Jung, J.H.; Lee, M.; Kim, D.W.; Park, M.W.; Choi, I.J.; Lee, J.H.; Lee, J.H.; Lee, S.R.; Lee, P.H., et al. Clinical Outcome of Rotational Atherectomy in Calcified Lesions in Korea-ROCK Registry. Medicina (Kaunas) 2021, 57.

Original text

During follow-up, patient management, including medical treatment, was performed in accordance with accepted guidelines and established standards of care. This study was approved by the local ethics committee of each hospital, and all patients provided written informed consent for the use of their clinical data for the registry study.

Modified text

During follow-up, patient management, including medical treatment such as peri-procedural anticoagulation and antiplatelet therapy, was performed in accordance with established standards of care and accepted guidelines.

Original text

The primary endpoint was the target-vessel failure (TVF), defined as cardiac death, target-vessel spontaneous myocardial infarction (TVMI), or target-vessel revascularization (TVR).

Modified text

The primary endpoint was the composite rate of target-vessel failure (TVF), defined as target-vessel revascularization (TVR), target-vessel spontaneous myocardial infarction (TVMI), or cardiac death.

Original text

The definition of outcomes were same as previously published report [15].

Modified text

The definition of outcomes were same as in the previously published report mentioned above [15].

  1. Lee, K.; Jung, J.H.; Lee, M.; Kim, D.W.; Park, M.W.; Choi, I.J.; Lee, J.H.; Lee, J.H.; Lee, S.R.; Lee, P.H., et al. Clinical Outcome of Rotational Atherectomy in Calcified Lesions in Korea-ROCK Registry. Medicina (Kaunas) 2021, 57.

Original text

newly diagnosed DM using the 2010 criteria of the American Diabetes Association.

Modified text

newly diagnosed DM by applying the 2010 criteria of the American Diabetes Association.

Original text

Peri-procedural MI was defined as a peak elevation of the creatine kinase-myocardial band >10-fold above the upper reference limit within 48 hours after the procedure.

Modified text

Peri-procedural MI was defined as a peak rise of the creatine kinase-myocardial band 10 times higher than the upper limit of normal within 48 hours after PCI.

Original text

Cerebrovascular accident (CVA) was defined as a focal neurological deficit of central origin lasting >24 h

Modified text

Cerebrovascular accident (CVA) was defined as a focal neurological defect of central origin lasting more than 24 hours

Original text

Chronic kidney disease (CKD) was defined as an estimated glomerular filtration rate < 60 mL/min/1.73 m2, as calculated using the Modification of Renal Diet (MDRD) equation from baseline serum creatinine

Modified text

Chronic kidney disease (CKD) was defined as a calculated glomerular filtration rate < 60 mL/min/1.73 m2 by the Modification of Renal Diet (MDRD) equation from baseline serum creatinine

Original text

Contrast-induced nephropathy (CIN) after PCI was defined as the impairment of kidney function, measured as either a 25% increase in serum creatinine from baseline or a 0.5 mg/dL increase in absolute serum creatinine value within 48-72 hours after the procedure.

Modified text

Contrast-induced nephropathy (CIN) after procedure was defined as impairment of the function of kidney, measured as either a 0.5 mg/dL rise in absolute serum creatinine level or a 25% rise in serum creatinine compared to baseline level within 48-72 hours after the procedure.

Original text

Categorical variables were expressed as numbers and percentages.

Modified text

Categorical variables were expressed by numbers and percentages.

Original text

Event rates were estimated using Kaplan–Meier estimates in time-to-first-event analyses and were compared using the log-rank test.

Modified text

Event rates were determined using Kaplan–Meier method in time-to-first-event analyses and were compared by the log-rank test.

Original text

A p value < 0.05 was considered statistically significant

Modified text

A p value < 0.05 was regarded statistically significant

Reviewer 2 Report

The study is very interesting and well-conducted. The authors evaluated the clinical impact of diabetes duration on clinical outcomes in patients with heavily calcified lesions that required rotational atherectomy during percutaneous coronary intervention. They demonstrated an association between a longer DM duration and a high incidence of target-vessel failure and repeat revascularization in patients with a longer duration of diabetes mellitus, compared with non-diabetes or shorter duration of diabetes mellitus. I consider that the study is valuable and has interesting findings that can have important implications for clinical practice. In order to improve the quality of the study, I have some suggestions:

1. The introduction is relatively short and the modifications must be made in accordance with what is outlined in the general observation, introducing the section on material and methods.

2. Could the total number of stents and their length, which were higher in the case of patients with diabetes, influence the results obtained?

Author Response

Response to Reviewer Comments

Point 1: The introduction is relatively short and the modifications must be made in accordance with what is outlined in the general observation, introducing the section on material and methods.

Response 1: I appreciated your valuable comments. As you mentioned, the content of the introduction seems to be relatively short. We reinforced the background by describing in more detail about relationship beetween DM duration and CAD. In addition, we briefly summarized method & material as reviewers comment.

Original text

Because DM is a chronic, progressive disease leading to micro- or macrovessel damage [7], the duration of diabetes (DM duration) is also associated with clinical outcomes in patients with CAD [8,9].

Modified text

Because DM is a chronic, progressive disease leading to micro- or macrovessel damage [7], the duration of diabetes (DM duration) is also associated with clinical outcomes in patients with CAD. In previous studies, DM duration was independently associated with cardiovascular mortality and increased the risk of coronary heart dis-ease death [8,9]. Since long period (> 10.5 years) of prevalence of DM is significantly associated with adverse cardiovascular events, it was suggested to consider routine CAD screening examination in these patients [10].

Original text

However, the clinical impact of DM duration is not well-known and there have been few studies describing the relationship between DM duration and clinical outcome af-ter PCI, especially in heavy calcified lesions. Therefore, we sought to determine the clinical impact of DM duration on procedural, in-hospital and mid-term clinical out-comes in patients with heavy CAC lesions underwent PCI using RA.

Modified text

However, the clinical impact of DM duration is not well-known and there have been no studies describing the relationship between DM duration and clinical outcome after PCI, especially in heavy calcified lesions. Therefore, we sought to determine the clini-cal impact of DM duration on clinical outcomes in patients with heavy CAC lesions underwent PCI using RA.

Based on the 10-year prevalence period, we compared the clinical outcome and procedure details during RA

Point 2: Could the total number of stents and their length, which were higher in the case of patients with diabetes, influence the results obtained?

Response 2: Thank you for your comment. Although the stent number and length were longer in the L-DM group than those in other groups, there was no statistically significant difference. Therefore, stent number or length could not make the difference for clinical outcome between DM vs. non-DM groups in our study

Reviewer 3 Report

Jung et al reports that the association between a longer diabetes mellites duration and poor clinical outcomes in patients with severe calcified CAD lesions. This report is well written and discussed their finding in detail with potential limitations of the study. Please address the following concerns.

General comments:

1.     In introduction, authors states, “there have been few studies describing the relationship between DM duration and clinical outcome after PCI, especially in heavy calcified lesions (line# 60-61)”. There is no reference to support this sentence.

2.     In material method section, authors describe “540 patients (583 lesions) with”. It is confusing. Please describe what is 540 and 583.

3.     Please add reference form “the 2010 criteria of the American Diabetes Association” line# 112

Author Response

Response to Reviewer Comments

Point 1: In introduction, authors states, “there have been few studies describing the relationship between DM duration and clinical outcome after PCI, especially in heavy calcified lesions (line# 60-61)”. There is no reference to support this sentence

Response 1: We appreciate your valuable comments. The sentence "line# 60-61" meant that “to the best of our knowledge there was no studies describing~”. The word “few studies” seems to be misleading, so I'll change it to a clearer sentence.

Original text

However, the clinical impact of DM duration is not well-known and there have been few studies describing the relationship between DM duration and clinical outcome af-ter PCI, especially in heavy calcified lesions

Modified text

However, the clinical impact of DM duration is not well-known and there have been no studies describing the relationship between DM duration and clinical outcome after PCI, especially in heavy calcified lesions.

Point 2: In material method section, authors describe “540 patients (583 lesions) with”. It is confusing. Please describe what is 540 and 583

Response 2: The number of lesions treated was 583 lesions because there were 540 patients enrolled in the registry and there were also patients with multi-vessel disease.

Point 3: Please add reference form “the 2010 criteria of the American Diabetes Association” line# 112

Response 3: As in your comment, omission of reference to that part has been confirmed. So sutable reference was added.

Modyfied text

Especially, diabetes was defined as either a previously diagnosed DM or newly diagnosed DM using the 2010 criteria of the American Diabetes Association. According to this definition, subjects with fasting glucose ≥126 mg/dl and/or glycated hemoglobin ≥6.5% and/or post-challenge glucose (glucose at 2 h after a 75 g oral glucose load) ≥200 mg/dl were newly diagnosed with DM [16].

  1. American Diabetes, A. Diagnosis and classification of diabetes mellitus. Diabetes Care 2010, 33 Suppl 1, S62-69.
